# miR-142 Targets TIM-1 in Human Endothelial Cells: Potential Implications for Stroke, COVID-19, Zika, Ebola, Dengue, and Other Viral Infections

**DOI:** 10.3390/ijms231810242

**Published:** 2022-09-06

**Authors:** Urna Kansakar, Jessica Gambardella, Fahimeh Varzideh, Roberta Avvisato, Stanislovas S. Jankauskas, Pasquale Mone, Alessandro Matarese, Gaetano Santulli

**Affiliations:** 1Department of Medicine, Division of Cardiology, Wilf Family Cardiovascular Research Institute, Einstein Institute for Aging Research, Albert Einstein College of Medicine, New York, NY 10461, USA; 2AORN “Antonio Cardarelli”, 80131 Naples, Italy; 3Department of Molecular Pharmacology, Institute for Neuroimmunology and Inflammation, Einstein-Mount Sinai Diabetes Research Center (ES-DRC), Fleischer Institute for Diabetes and Metabolism (FIDAM), Albert Einstein College of Medicine, New York, NY 10461, USA

**Keywords:** blood–brain barrier, cerebrovascular disease, Chikungunya virus, COVID-19, endothelial cells, HAVCR-1, hBMECs, Japanese encephalitis virus, KIM-1, Lassa virus, Marburg virus, microRNA, miR-142-3p, SARS-CoV-2, stroke, West Nile virus

## Abstract

T-cell immunoglobulin and mucin domain 1 (TIM-1) has been recently identified as one of the factors involved in the internalization of the severe acute respiratory syndrome coronavirus 2 (SARS-CoV-2) in human cells, in addition to angiotensin-converting enzyme 2 (ACE2), transmembrane serine protease 2 (TMPRSS2), neuropilin-1, and others. We hypothesized that specific microRNAs could target TIM-1, with potential implications for the management of patients suffering from coronavirus disease 2019 (COVID-19). By combining bioinformatic analyses and functional assays, we identified miR-142 as a specific regulator of TIM-1 transcription. Since TIM-1 has been implicated in the regulation of endothelial function at the level of the blood-brain barrier (BBB) and its levels have been shown to be associated with stroke and cerebral ischemia-reperfusion injury, we validated miR-142 as a functional modulator of TIM-1 in human brain microvascular endothelial cells (hBMECs). Taken together, our results indicate that miR-142 targets TIM-1, representing a novel strategy against cerebrovascular disorders, as well as systemic complications of SARS-CoV-2 and other viral infections.

## 1. Introduction

MicroRNAs (miRNAs, miRs) are a group of small (18–24 nucleotides) non-coding RNAs that control the post-transcriptional expression of target genes [1,2,3,4]. Specifically, miRNAs are able to recognize and bind the 3′ untranslated region (UTR) of certain genes, inhibiting their expression. Therefore, miRNAs are involved in a number of physiological and pathological processes [5,6,7,8,9,10,11,12,13,14]. We and others have identified several miRNAs involved in the regulation of endothelial function [15,16,17,18,19,20].

T-cell immunoglobulin and mucin domain 1 (TIM-1) is a type I cell-surface glycoprotein that includes four main domains: a carboxy-terminal cytoplasmic tail, a mucin domain, a single transmembrane domain, and an amino-terminal immunoglobulin (Ig)-like domain [21]. TIM-1 is expressed on endothelial cells, antigen-presenting cells, neurons and microglial cells, and proximal tubular cells in the kidney (it is also known as Kidney Injury Molecule-1; KIM-1).

TIM-1 has been shown to act as a cell receptor or entry factor for a number of viruses, including Hepatitis A (indeed, another of its nomenclatures is Hepatitis A Virus Cellular Receptor 1; HAVcR1) [22,23], Ebola virus [24,25,26], West Nile virus [26], Lassa virus [27], Marburg virus [28], Dengue virus [29,30], Japanese encephalitis virus [31], Chikungunya virus [32], and Zika virus [33,34]. Intriguingly, emerging investigations have proposed TIM-1 as a co-factor for the internalization of SARS-CoV-2 in human cells [35,36,37]. Specifically, TIM-1 has been shown to enhance a recombinant replication-competent vesicular stomatitis virus that encodes SARS-CoV-2 spike (rVSV/Spike) infection over a wide range of ACE2 concentrations [38], whereas a mutant of TIM-1 that does not bind to phosphatidylserine was unable to facilitate SARS-CoV-2 entry [39].

Since TIM-1 has been implicated in the regulation of endothelial function at the level of the blood–brain barrier (BBB), and its levels have been shown to be associated with stroke and cerebral ischemia-reperfusion injury [21,40], the main aim of this study was to identify and validate miRNAs that specifically target TIM-1 in human brain microvascular endothelial cells (hBMECs).

## 2. Results

### 2.1. miR-142 Targets TIM-1 in Different Species

By combining bioinformatic analyses and functional assays, we identified hsa-miR-142-3p (abbreviated as miR-142) as a specific and conserved regulator of TIM-1 transcription (Figure 1). We also generated a mutant construct of TIM-1 3′-UTR (TIM-1 MUT), harboring nucleotide substitutions within the predicted miR-142 binding sites of TIM-1 3′-UTR (Figure 1).

### 2.2. Validation in Human Endothelial Cells of the Transcriptional Regulation of TIM-1 by miR-142

Preclinical assays have shown that inhibiting TIM-1 protects against cerebral ischemia-reperfusion injury [21], while in a clinical study conducted on 4591 subjects, TIM-1 has been associated with a significantly increased incidence of stroke (both ischemic stroke and all-cause stroke) [40]. On these grounds, we validated, for the first time to the best of our knowledge, miR-142 as a functional modulator of TIM-1 in hBMECs, which are generally considered to be among the most suitable cell lines to recapitulate the human BBB in vitro [41,42].

We demonstrated via luciferase assays (Figure 2) that TIM-1 is a specific molecular target of miR-142.

### 2.3. TIM-1 Expression Levels Are Regulated by miR-142

We observed that miR-142 was able to reduce both the mRNA levels (Figure 3; Table 1) and the protein levels (Figure 4) of TIM-1.

We also confirmed that TIM-1 was expressed in other endothelial cell types, namely adult human lung microvascular endothelial cells (HMVEC-Ls) and human umbilical vein endothelial cells (HUVECs), demonstrating that its expression was significantly reduced by miR-142 (Appendix A).

### 2.4. miR-142 Attenuates TIM-1-Induced Endothelial Permeability

An increased endothelial leakage represents a pathophysiological hallmark of endothelial dysfunction, especially in the settings of viral infections, including COVID-19 [43,44,45,46,47,48,49,50].

In our experimental setting, we demonstrated that miR-142 significantly attenuated endothelial permeability triggered by the main agonist of TIM-1, namely, TIMD4 [51,52] (Figure 5).

Our data on endothelial leakage were further supported by the regulation of a major tight-junction protein, namely occludin [53,54,55,56], by miR-142 (Appendix A).

## 3. Discussion

Our data indicate that miR-142 targets TIM-1, representing a novel potential strategy against cerebrovascular diseases, COVID-19, and other viral infections.

We established that TIM-1 is expressed in different types of endothelial cells and that miR-142 is able to significantly reduce its expression. One of the key findings of the present work is the demonstration of the functional role of miR-142 in the regulation of endothelial leakage, which represents a fundamental step in several disorders caused by different viruses, which do not have to necessarily display a definite endotheliotropism [57,58,59,60,61,62,63,64]. Moreover, many viral infections, including COVID-19, have been shown to lead to the involvement of different tissues and organs, often culminating in a systemic inflammatory response, in which the endothelium plays key roles [50,65,66,67,68]. Our results in terms of endothelial permeability are also corroborated by data showing that miR-142 can regulate the expression of Occludin, a tight-junction protein that is functionally involved in viral neuro-invasion [69].

Since TIM-1 has been shown to represent an entry co-factor for a number of viruses causing major diseases in humans [22,23,24,25,26,27,28,29,30,31,32,33,34,35,36,37,39,70], our findings have a major relevance in terms of public health and should enthuse further dedicated investigations exploiting miR-142 as a biomarker as well as a potential target for novel therapeutic approaches. For instance, in line with these observations, we have previously demonstrated that endothelial miRNAs can be harnessed as reliable biomarkers for cerebrovascular complications of COVID-19 [71]. Notably, the crucial role of endothelial dysfunction in the pathobiology of COVID-19, initially described by our research group [72] has been later confirmed by numerous investigators [72,73,74,75,76,77,78,79,80,81,82,83,84]. Indeed, endothelial cells have been shown to express the key co-factors involved for the internalization of SARS-CoV-2 in host cells, including angiotensin converting enzyme 2 (ACE2), transmembrane protease serine 2 (TMPRSS2), cathepsins B and D, TIM-1, neuropilin-1, and others, thereby representing a natural target of SARS-CoV-2 [83,85,86,87,88,89,90,91,92]. Furthermore, the systemic inflammatory viral reaction observed in patients affected by COVID-19 has been shown to be linked to endothelial dysfunction [93,94,95,96,97], leading to thromboembolic events [98,99,100], which represent a common feature of COVID-19 cases with a severe outcome [101,102,103].

Additional potential applications of our discovery include kidney disease, disturbances of iron metabolism, and the modulation of the immune response. Indeed, plasma levels of TIM-1 have been found to be associated with underlying tubulointerstitial and mesangial lesions and progression to kidney failure in two cohort studies of individuals with kidney diseases [104]. Preclinical investigations have demonstrated that TIM-2, the rodent homolog to TIM-1, is a binding partner to H-ferritin [105], a protein initially thought to be solely used for iron storage [106,107,108], which was later shown to serve as an iron delivery protein, secreted from the endothelial cells within the BBB as a source of iron for the brain [109]. TIM-1 can also mediate the tethering between T cells and endothelial cells in vivo and the rolling of lymphocytes on the vascular endothelium [110].

Our study is not exempt from limitations, including having performed the luciferase assays only in one cell type; nevertheless, we demonstrated that TIM-1 is indeed expressed (and regulated by miR-142) in three different types of endothelial cells, namely, hBMECs, HUVECs, and HMVEC-Ls. Further dedicated experiments are warranted to prove the effects of miR-142 in the pathophysiology of cerebrovascular events and on the actions of SARS-CoV-2 and other viruses.

## 4. Materials and Methods

### 4.1. Cell Culture and Reagents

All reagents were purchased from Millipore-Sigma (Burlington, MA, USA), unless otherwise stated. We obtained hBMECs from Neuromics (Minneapolis, MN, USA; #HEC02) [111]; these cells have been proved to be the most suitable human cell line to reproduce the BBB in vitro [112]. Adult human lung microvascular endothelial cells (HMVEC-Ls) were obtained from Lonza (Basel, Switzerland; catalog number, CC-2527) and human umbilical vein endothelial cells (HUVECs) from ThermoFisher Scientific (Waltham, MA, USA; Catalog number, #C0035C). Cells were cultured in a standard humidified atmosphere (37 °C) containing 5% CO_2_, as we previously described [111,113]. In some experiments, cells were transfected with *pcDNA3.1-TIM-1* plasmids (GenScript, Piscataway, NJ, USA).

### 4.2. Identification and Validation of miR-142 as a Regulator of TIM-1

To identify miRNAs targeting the 3′-UTR of TIM-1, we used TargetScanHuman 8.0, as we previously described [20,111,114,115,116,117]. To assess the effects of miR-142 on TIM-1 gene transcription, we used a luciferase reporter containing the 3′-UTR of the predicted miRNA interaction site, both wild-type and mutated, in hBMECs cells. The mutant construct of TIM-1 3′-UTR (TIM-1 MUT, as shown in Figure 1), harboring a substitution of three nucleotides within the predicted miR-142 binding sites of TIM-1 3′-UTR was obtained using the NEBaseChanger and Q5 site-directed mutagenesis kit (New England Biolabs, Ipswich, MA, USA) as we previously described [111,114,116].

We transfected hBMECs with the 3′-UTR reporter plasmid (0.05 μg) and miR-142 mimic (ThermoFisher Scientific, Waltham MA, USA) or miR-142 inhibitor, as well as a non-targeting negative control (scramble), all used at a final concentration of 50 nMol/L, using Lipofectamine RNAiMAX (ThermoFisher Scientific) [111,116]. Firefly and Renilla luciferase activities were measured 48 h after transfection, using Luciferase Reporter Assay System (Promega, Madison, WI, USA), normalizing Firefly luciferase to Renilla luciferase activity [20,111,114]. The cellular expression of TIM-1 was determined using RT-qPCR as we previously described [20,114,117,118], normalizing to glyceraldehyde 3-phosphate dehydrogenase (GAPDH). The sequences of oligonucleotide primers (Merck, Darmstadt, Germany) are reported in Table 1.

### 4.3. Immunoblotting

Immunoblotting assays were performed as previously described and validated by our group [111,116,119]; the intensity of the bands was quantified using FIJI (“Fiji Is Just ImageJ”) software. The antibody for TIM-1 was purchased from Novus Biologicals (Centennial, CO, USA; catalog number; #NBP1-76701); the antibody for Occludin was purchased from Novus Biologicals (catalog number, #NBP1-87402); the antibody for β-Actin was purchased from abcam (Cambridge, MA, USA; catalog number, #ab8229); the antibody for GAPDH was purchased from Novus Biologicals (catalog number, #NB300-221).

### 4.4. Endothelial Permeability Assay

We performed the in vitro permeability assay in hBMECs as we previously described [53,111,113]. Briefly, hBMECs transfected with miR-142 mimic or miR scramble were grown on 0.4 mm fibronectin-coated transwell filters for 48 h; then, we replaced the medium in the upper well with FITC-dextran 70 kD (0.5 mg/mL in phosphate-buffered saline, PBS). Cells were stimulated in the lower well with PBS alone or PBS containing 50 ng/mL VEGF-A_165_ (Bio-Techne Corporation, Minneapolis, MN, USA). Endothelial permeabilization was quantified by measuring at 520 nm the fluorescence of dextran that passed in the bottom chamber through the cell monolayer [111,113,120].

### 4.5. Statistical Analysis

All data were expressed as means ± standard errors of the means (SEMs). The statistical analyses were carried out using GraphPad 9 (Prism, San Diego, CA, USA). Statistical significance, set at *p* < 0.05, was tested using the two-way ANOVA followed by Tukey–Kramer multiple comparison test or the non-parametric Mann–Whitney U test, as appropriate.

## 5. Conclusions

Taken together, our results indicate that miR-142 targets TIM-1, representing a novel potential strategy against cerebrovascular disease, SARS-CoV-2, and other viral infections.

## Figures and Tables

**Figure 1 ijms-23-10242-f001:**
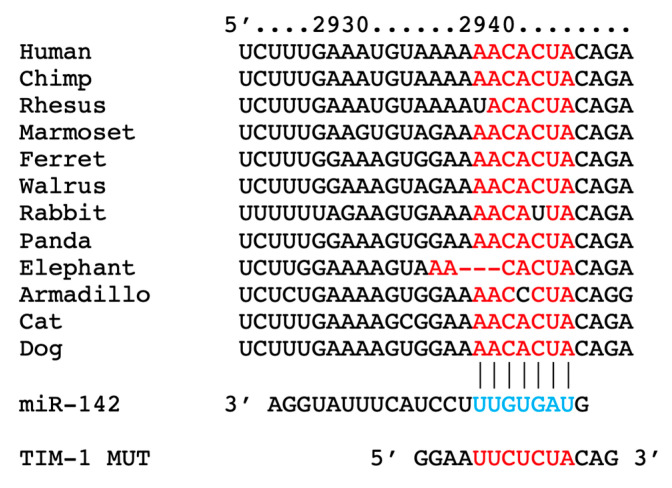
Identification of miR-142 as a specific modulator of TIM-1; the complementary nucleotides between the target region of TIM-1 3′-UTR and hsa-miR-142-3p are highly conserved across different species.

**Figure 2 ijms-23-10242-f002:**
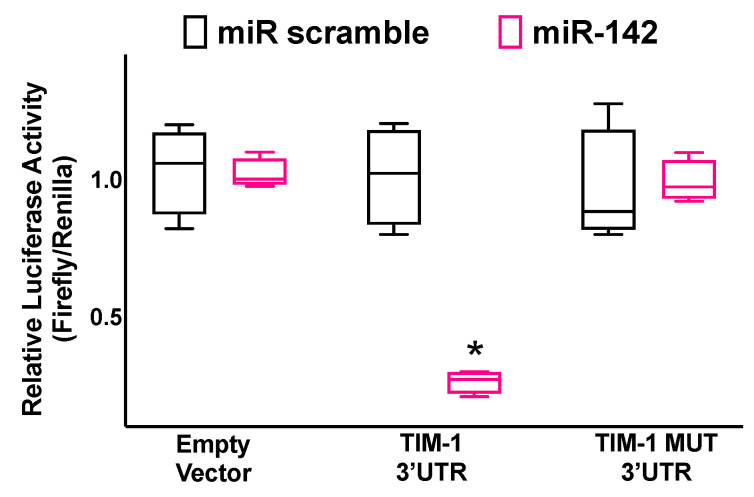
Validation of TIM-1 targeting by miR-142. Luciferase activity was measured in hBMECs 48 h after transfection, using the vector without TIM-1 3′-UTR (empty vector), the vector containing the wild-type TIM-1 3′-UTR, and the vector containing a mutated TIM-1 3′-UTR (TIM-1 MUT); a non-targeting miRNA (miR scramble) was employed as a further control. All experiments were performed at least in triplicate; the box-and-whiskers graph indicates the medians and the 5th–95th percentiles; * *p* < 0.01 vs. miR scramble.

**Figure 3 ijms-23-10242-f003:**
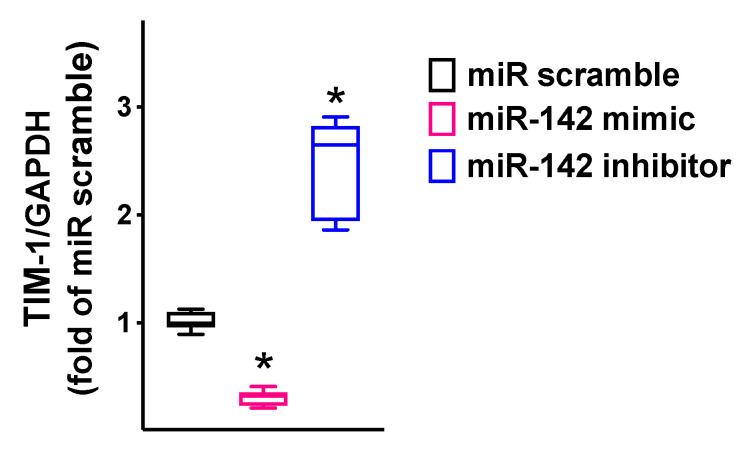
TIM-1 expression in hBMECs was reduced by miR-142 and increased by miR-142 inhibitor. TIM-1 mRNA levels were measured using RT-qPCR in hBMECs transfected with miR-142 mimic, inhibitor, or scramble (negative control) for 48 h, normalizing to glyceraldehyde 3-phosphate dehydrogenase (GAPDH). All experiments were performed at least in triplicate; the box-and-whiskers graph shows the medians and the 5th–95th percentiles; * *p* < 0.01 vs. miR scramble. Sequences of oligonucleotide primers are reported in Table 1.

**Figure 4 ijms-23-10242-f004:**
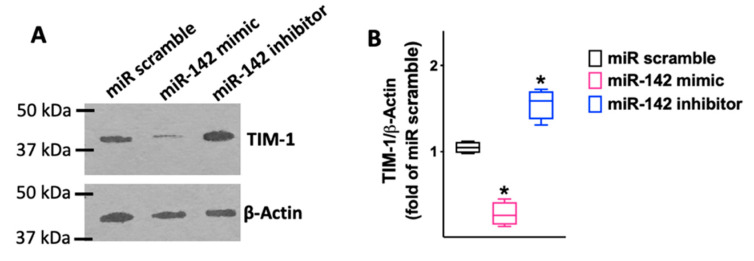
The results obtained via RT-qPCR were confirmed in terms of protein levels (*Biorbyt*; #orb382613; Cambridge, UK), as shown in the representative immunoblots in panel (**A**) and their quantification in panel (**B**). All experiments were performed at least in triplicate; the box-and-whiskers graph indicates the medians and the 5th–95th percentiles; *: *p* < 0.01 vs. miR scramble.

**Figure 5 ijms-23-10242-f005:**
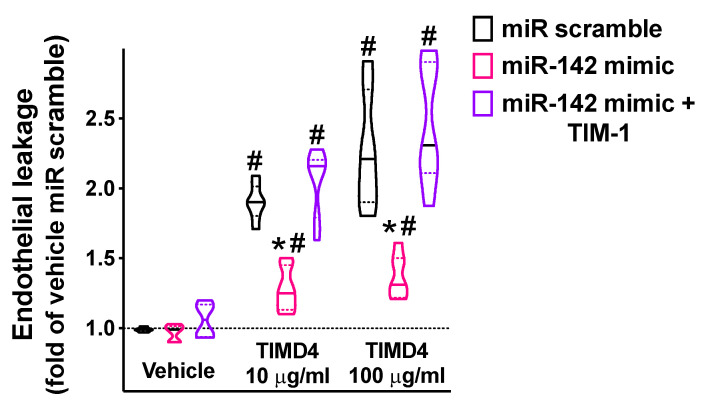
The targeting of TIM-1 by miR-142 significantly reduced endothelial permeability. Endothelial leakage triggered by 48 h incubation with TIM-1 agonist TIMD4 (10 or 100 ng/mL; *BioLegend* San Diego, CA, USA) was measured in hBMECs transfected with miR-142 mimic, miR scramble, or by combining miR-142 mimic and TIM-1 overexpression (pcDNA3.1-TIM-1 plasmid; *GenScript*, Piscataway, NJ, USA). All experiments were performed at least in triplicate; in the violin plot, median (solid line) and quartiles (dotted lines) are indicated; * *p* < 0.01 vs. miR scramble, # *p* < 0.01 vs. vehicle.

**Table 1 ijms-23-10242-t001:** Oligonucleotide primer sequences.

Gene	Primer	Sequence (5′-3′)	Amplicon (bp)
**TIM-1**	** *Forward* **	CAT AAA CCT GGC CTA CGT GC	99
	** *Reverse* **	AGA GGG TCA GCA GGA AAA CA	
**GAPDH**	** *Forward* **	GGC TCC CTT GGG TAT ATG GT	94
	** *Reverse* **	TTG ATT TTG GAG GGA TCT CG	

bp, base pairs; GAPDH, glyceraldehyde 3-phosphate dehydrogenase.

## Data Availability

Data supporting the reported results are contained within this article.

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
