# Peer review of "miR-142 Targets TIM-1 in Human Endothelial Cells: Potential Implications for Stroke, COVID-19, Zika, Ebola, Dengue, and Other Viral Infections"

_ijms, 2022, doi:10.3390/ijms231810242_

Round 1

Reviewer 1 Report

The submission by Kansakar, et al is a very interesting one, providing new insights on the pathogenesis of endothelial function and permeability with regards to the role a specific micro RNA (miR-142) plays. On the basis of these considerations, the authors hypothesizes that specific microRNA miR-142 could target TIM-1, and in turn, considering the role of endothelium on the pathogenesis of SARS-CoV2 infection, they suggest that miR-142 may have major implications in the management of patients withCOVID-19.

However, for a nonspecialized reader, there is a gap between the results the study shows, and the potential implications largely developed throughout the Discusssion Section, and definitely accepted in their conclusion.

Indeed, in their Results Section, the authors show that :

-           TIM-1 is a specific molecular 70 target of miR-142.

-          TIM-1 expression levels are regulated by miR-142

-          miR-142 attenuates TIM-1-induced endothelial permeability.

Also, in their discussion and conclusion, the authors suggest « our data indicate that miR-142 targets TIM-1, representing a novel potential strategy 121 against COVID-19 ».

As this is uneasy to understand by a nonspecialized reader, it could be useful to explain the process exposed step by step.

Author Response

R: We thank this Reviewer for her/his suggestion. In the revised version of the manuscript, we have better explained the pathophysiology linking miR-142, TIM-1 and endothelial permeability, as requested; we have also expanded the introduction on miRNAs, to make the paper easier to understand by non-specialized Readers, as pertinently recommended by this Reviewer.

Reviewer 2 Report

Santulli et al describe the results indicate that miR-142 targets TIM-1, representing a novel potential strategy against SARS-CoV-2 and other viral infections. I recommend its publication in the IJMS, pending the following reference.

Provide the reference ACS Infect. Dis. 2020, 6, 11, 2844–2865 in the introduction section.

Author Response

R: We thank this Reviewer for her/his comment. The suggested report has now been cited.

Reviewer 3 Report

In this manuscript, Kansakar et al. have identified a novel microRNA, miR-142, that is altering TIM-1 expression in hBMECs. By using qPCR, WB, and luciferase assay, authors indicated that miR-142 is downregulating TIM-1 in both transcription and translation level. Moreover, this interruption leads to the inhibition of endothelial leakage triggered by TIM-1.

The concept of this manuscript is straightforward and results are quite clear, means the sensitivity and specificity of miR-142 is ideal to interfere TIM-1 expression, at least in hBMECs. However, even though authors claimed that hBMECs are one of the most suitable models for such kind of study (in BBB), including another cell line / primary culture / stem cell-derived model could be more convincing. What’s the expression level of TIM-1 in other cell types? Is the expression of TIM-1 universal in different organ-derived cells?

The descriptions in the Results part seems over-simplified, which includes the labels in figures and figure legends. There is many information only mentioned in material and method, which is in the later part of the manuscript, such as TIM-1 MUT in Figure 1. (Line 56).

In the discussion, authors have cited many publications to emphasize the importance of TIM-1, but details are lacking. For example, from Line 128 to 138, there are 15 publications been mentioned, but the points are blur and descriptions are weak.

Authors tried to link the TIM-1 inhibition by miR-142 to SARS-CoV-2 infection and claimed that this can be a potential angle to cure COVID-19. But evidences are still weak. Firstly, TIM-1 is not necessary for SARS-CoV-2 infection. From the two citations in the manuscript, TIM-1 is only a potential host factor that may have some contributions in particular model of SARS-CoV-2 infection. Secondly, this manuscript used huge portions in the title, introduction, discussion, and even conclusion to described SARS-CoV-2 and COVID-19, but there is no direct experiments or results that provide evidence of miR-142 may alter SARS-CoV-2, or even other BSL2 coronaviruses, infection. This could be a misleading to the readers

Minor correction: There are two Figure 4s and in the second Figure 4, the concentrations of TIMD4 are inconsistent with the description in figure legend.

Author Response

In this manuscript, Kansakar et al. have identified a novel microRNA, miR-142, that is altering TIM-1 expression in hBMECs. By using qPCR, WB, and luciferase assay, authors indicated that miR-142 is downregulating TIM-1 in both transcription and translation level. Moreover, this interruption leads to the inhibition of endothelial leakage triggered by TIM-1.

The concept of this manuscript is straightforward and results are quite clear, means the sensitivity and specificity of miR-142 is ideal to interfere TIM-1 expression, at least in hBMECs.

R: We thank this Reviewer for the words of appreciation toward our work.

However, even though authors claimed that hBMECs are one of the most suitable models for such kind of study (in BBB), including another cell line / primary culture / stem cell-derived model could be more convincing. What’s the expression level of TIM-1 in other cell types? Is the expression of TIM-1 universal in different organ-derived cells?

R: We thank this Reviewer for this insightful remark. We agree that showing TIM-1 expression in just one type of endothelial cells was somehow reductive. However, we had clearly specified that since TIM-1 had been implicated in the regulation of endothelial function at the level of the blood-brain barrier (BBB), and its levels had been shown to be associated with stroke and cerebral ischemia-reperfusion injury, the main aim of our study was to identify miRNAs that specifically target TIM-1 in hBMECs.

Nevertheless, following your suggestion, in the revised version of the manuscript, we show that TIM-1 is indeed expressed in other cell types, namely HUVECs, and human lung microvascular endothelial cells (HMVEC-L).

The descriptions in the Results part seems over-simplified, which includes the labels in figures and figure legends. There is many information only mentioned in material and method, which is in the later part of the manuscript, such as TIM-1 MUT in Figure 1. (Line 56).

In the discussion, authors have cited many publications to emphasize the importance of TIM-1, but details are lacking. For example, from Line 128 to 138, there are 15 publications been mentioned, but the points are blur and descriptions are weak.

R: We have expanded the description of the Results, as well as the Discussion, as suggested by this Reviewer.

Authors tried to link the TIM-1 inhibition by miR-142 to SARS-CoV-2 infection and claimed that this can be a potential angle to cure COVID-19. But evidences are still weak. Firstly, TIM-1 is not necessary for SARS-CoV-2 infection. From the two citations in the manuscript, TIM-1 is only a potential host factor that may have some contributions in particular model of SARS-CoV-2 infection. Secondly, this manuscript used huge portions in the title, introduction, discussion, and even conclusion to described SARS-CoV-2 and COVID-19, but there is no direct experiments or results that provide evidence of miR-142 may alter SARS-CoV-2, or even other BSL2 coronaviruses, infection. This could be a misleading to the readers.

R: We again agree with this Reviewer: therefore, in the revised manuscript, we have toned down these statements and discussed these aspects in the limitations of the study.

Minor correction: There are two Figure 4s and in the second Figure 4, the concentrations of TIMD4 are inconsistent with the description in figure legend.

R: Rectified; thanks!

Reviewer 4 Report

The paper has serious fall and should not be accepted in the present form and I suggest rejection. Indeed, in my opinion, the introduction is too short and the aim of the study is foggy.    The materials and methods are fine but the results are very short and not explain the data obtained.    Furthermore, the manuscript has some gaps in the results section since the Authors do not explain how the sars cov 2 can cross the blood brain barrier (directly or indirectly, maybe recruiting immune system cells and let they to release cytokines as previously hypothesized. Furthermore, it should be fine if the Authors showed some tight junction decrease or dislocation both by western blotting analysis or by immuno fluorescence staining in order to reinforce their results.    Finally, the discussion section is very short (as the results and the introduction section). More experiments are needed before the paper should be considered for publication.

Author Response

"The paper has serious fall and should not be accepted in the present
form and I suggest rejection. Indeed, in my opinion, the introduction is
too short and the aim of the study is foggy.
R: The introduction has been extended, as requested

We respectfully disagree on the fact that the aim of the study is foggy: “the main aim of this study was to identify miRNAs that specifically target TIM-1 in human brain microvascular endothelial cells (hBMECs)”.

The materials and methods are fine but the results are very short and
not explain the data obtained.

R: The results have been extended, and the data obtained are now better explained.

Furthermore, the manuscript has some gaps in the results section since
the Authors do not explain how the sars cov 2 can cross the blood brain
barrier (directly or indirectly, maybe recruiting immune system cells
and let they to release cytokines as previously hypothesized.
R: We respectfully want to point out that these points are beyond the aim of the paper. In the revised version of the manuscript, we toned down the conclusions, clearly specifying that our findings (miR-142 targets TIM-1 in hBMECs) have potential implications for several viral infections, including SARS-CoV-2. We also clarify in the title that the implications for viral infections are potential.

Furthermore, it should be fine if the Authors showed some tight junction
decrease or dislocation both by western blotting analysis or by immuno
fluorescence staining in order to reinforce their results.
R: We thank this Reviewer for her/his pertinent suggestion. In the revised manuscript, we now show by western blot (immunofluorescence is not a quantitative assay) that miR-142 regulates the expression of Occludin, as requested.

Finally, the discussion section is very short (as the results and the
introduction section). More experiments are needed before the paper
should be considered for publication.

R: We have expanded introduction, results, and discussion, as requested. The paper is now 17 pages.

Round 2

Reviewer 3 Report

The authors have provided more results from different cell types to strengthen the effect of miR-142 on repressing TIM-1.

Although more references have been mentioned, the anti-viral potential mentioned in this article is somehow still not persuasive. Authors shouldn't put too much weight on that direction. On the other hand, over-citation is also a noticeable issue in this manuscript. These things make the manuscript seems overly ambitious and empty. More contents and data are needed.

One virological point needed to be made: authors need to be careful about using the term "necessary" in virus-host interaction. Bohan et al in their PLoS Pathogens paper (https://pubmed.ncbi.nlm.nih.gov/34797899/) has mentioned that without ACE2, expressing TIM-1 alone did not mediate infection. Which means TIM-1, like TMPRSS2 and CTS-L, is not a "necessary gene" in SARS-CoV-2 infection. Authors need to understand the differences to make such claim. 

Another minor point, there is no description of Western blot in Material and Method.

Author Response

R: We fully agree with this Reviewer and we have toned down the importance of TIM-1 in COVID-19 (we removed the word “necessary”), mentioning the Bohan’s paper suggested by this Reviewer. Western blot methodology has been added; thanks.

Reviewer 4 Report

Dear Authors, 

First of all thanks for your replay and for editing you manuscript. 

However, even if the paper has been improved, the data obtained are not so exhaustive for supporting your conclusion.

1-The paper now show one experimet performed on other endothelial cells (HMVEC-L and HUVEC). There are evidence that miR-142 reduce, for example, endothelial permeability as reported for brain endothelial cells?

2-The occludin western blotting is fine, but I suggest you to add this kind of information also in the main manuscript.

3- The Authors stated that the miR-142 strategy should be fine for different viral infection, even if they underline the covid-19 infection. it should be better to elucidate this point. Indeed, different viruses can enter the cells by other ways and not only by TIM-1. Please, better elucidate this in discussion section.

4- the Authors sometimes mention the SARS-CoV-2 complications. On this regards, as I report in point 1, it should be better to perform experiments on lung endothelial cells (HMVEC-L). 

5- If the aim of the study is to "identify miRNAs that specifically target TIM-1 in human brain microvascular endothelial cells (hBMECs)", why in the title you mention general endothelial cells? And in the Materials and Methods and Results you quote experiments on other endothelial cell line?

6- The Authors indicate that the levels of TIM-1 is associated with stroke and cerebral ischemia-reperfusion injury. Did the permformed experiments showing the potential role of miR-142 on this regard? What about the relation between stroke, ischemia, covid-19 internalization, severe acute respiratory syndrome? 

7-No explanations on immunoblot analysis was added in the materials and methods section. Please add.

Author Response

R: We thank this Reviewer for her/his additional comments. As requested by this Reviewer, we now specify in the title that we are focusing on in human brain microvascular endothelial cells; we performed additional experiments on other endothelial cell types (including HMVEC-L) simply because we were asked to do so by one of the Reviewers, who wanted to see if there was any effect in other cell types. The association between TIM-1 and stroke has been demonstrated in “Arterioscler Thromb Vasc Biol 2020;40:1777-1786”; miR-142 was not investigated in that study. We clarify in the discussion that further investigations on the relation between stroke, ischemia, covid-19 internalization, and severe acute respiratory syndrome are needed. Immunoblot methodology has been added; thanks.